# Fast electrical modulation of strong near-field interactions between erbium emitters and graphene

Daniel Cano [1], Alban Ferrier [2,3], Karuppasamy Soundarapandian[1], Antoine Reserbat-Plantey[1],
Marion Scarafagio[2], Alexandre Tallaire[2], Antoine Seyeux[2], Philippe Marcus[2], Hugues de Riedmatten[1,4],
Philippe Goldner[2], Frank H. L. Koppens [1,4✉] & Klaas-Jan Tielrooij [5✉]

Combining the quantum optical properties of single-photon emitters with the strong near-field interactions available in nanophotonic and plasmonic systems is a powerful way of creating quantum manipulation and metrological functionalities. The ability to actively and dynamically modulate emitter-environment interactions is of particular interest in this regard. While thermal, mechanical and optical modulation have been demonstrated, electrical modulation has remained an outstanding challenge. Here we realize fast, all-electrical modulation of the near-field interactions between a nanolayer of erbium emitters and graphene, by in-situ tuning the Fermi energy of graphene. We demonstrate strong interactions with a >1000-fold increased decay rate for ~25% of the emitters, and electrically modulate these interactions with frequencies up to 300 kHz – orders of magnitude faster than the emitter's radiative decay (~100 Hz). This constitutes an enabling platform for integrated quantum technologies, opening routes to quantum entanglement generation by collective plasmon emission or photon emission with controlled waveform.

[1] ICFO – Institut de Ciències Fotòniques, The Barcelona Institute of Science and Technology, 08860 Castelldefels (Barcelona), Spain. [2] Institut de Recherche de Chimie Paris (IRCP), Université PSL, Chimie ParisTech, CNRS, 75005 Paris, France. [3] Faculté des Sciences et Ingénierie, Sorbonne Universités, UFR 933, 75005 Paris, France. [4] ICREA - Institució Catalana de Reçerca i Estudis Avancats, 08010 Barcelona, Spain. [5] Catalan Institute of Nanoscience and Nanotechnology (ICN2), BIST and CSIC, Campus UAB, 08193 Bellaterra (Barcelona), Spain. ✉email: frank.koppens@icfo.eu; klaas.tielrooij@icn2.cat

One of the major accomplishments of nanophotonics research is the accurate control of the near-field inter-actions between light emitters and their environment. These interactions typically lead to a modification of the decay rate and emission properties, with decay-enhancement factors $F_P$ up to 1000 for emitters in the near-field of metallic nano-antennas[1]. Such systems, however, do not easily allow for active modulation of the emitter–environment interactions. Active and dynamic control of emitter–environment interactions has been achieved in photonic crystal cavities and nano-electromechanical systems through modulation of the effective refractive index of the environment, using electromechanical actuation[2], mechanical oscillations[3,4], or photoexcitation of free carriers[5]. These techniques, however, either only work with one fixed modulation frequency or require optical fields for fast modulation. Furthermore, the achieved decay-enhancements are typically moderate, with $F_P \sim 10$.

An effective solution can be found in graphene, a two-dimensional (2D) material that combines a high degree of electro-optical tunability at high speeds (tens of GHz[6]) with strong near-field light–matter interactions. Graphene not only provides a wide range of electrical tunability through its Fermi energy. At suffi-ciently high Fermi energy, it supports plasmons with much stronger field confinement than noble metals: the wavelength of graphene plasmons $\lambda_{pl}$ is ~100 times smaller than that of free-space photons $\lambda_0$, whereas in metallic thin films $\lambda_0/\lambda_{pl}$ is typically <3, see ref. [7]. For pristine graphene, this leads to a mode volume confinement of $(\lambda_0/\lambda_{pl})^3 \simeq 10^6$, according to ref. [8]. Moreover, it is possible to reach a mode volume confinement up to $10^9$ in graphene-metal grating structures[9] and $10^{10}$ in cube-grating structures[10], with potential for very strong near-field interactions with nearby emitters. Many suggested plasmon-based technolo-gies, where the fields of photonics and electronics merge[11,12], would benefit from fast temporal control of emitter–plasmon interaction, including applications such as single-photon nano-antennas[13], and sub-diffraction-limited sensors[14]. Experimen-tally, however, active electrical tunability of emitter–graphene interaction has only been demonstrated with low decay-enhancement ($F_P < 3$) and in a non-dynamic fashion[15,16].

Here we present a hybrid system made of a nanoscale layer of erbium emitters in an oxide matrix, and graphene, in which the near-field interactions can be efficiently controlled on-chip, at high modulation frequencies, and by means of moderate electrical gate voltages (on the order of a few volt). Rare-earth erbium ions are technologically highly relevant as they emit photons at 1.54-μm wavelength, within the C-band of optical communication systems. Furthermore, rare-earth oxide crystals have a proven relevance in photonic quantum memories[17] and spin–photon interfaces[18], while nanoscale rare-earth materials are currently gaining interest[19]. We will show highly efficient erbium–graphene interaction with decay-enhancement factors $F_P$ >1000 for ~25% of the ions, which means that >99.9% of the energy of these excited ions flows to graphene through near-field interactions. Importantly, we demonstrate modulation of the erbium–graphene interaction on a much shorter time scale than the lifetime of the single-photon emitters. In this special dynamical case, the quantum regime emerges, with exciting possibilities such as generation of single photons with controlled waveform[20,21], Dicke phases[22,23], and quantum entanglement generation by collective plasmon emission[24].

## Results
### Hybrid erbium–graphene system with dual-gate modulation.
We show the design of our hybrid system schematically in Fig. 1. The central part consists of a graphene monolayer on a nanoscale layer of erbium-doped $Y_2O_3$ (2%). To achieve strong near-field

interactions, the erbium emitters should be located at nanoscale distances from the 2D material—ideally within the sub-wavelength volume occupied by the highly confined plasmons (<15 nm). A sufficiently thin layer with erbium emitters is therefore required. However, crystals of nanoscale dimensions typically suffer from detrimental non-radiative losses due to surface defects[25]. Removing this loss mechanism is crucial, as it constitutes a competing energy flow channel for erbium–graphene interactions that would hinder the observation of these interactions. We overcome this experimental bottleneck by using atomic layer deposition (ALD)[26] with optimized post-treatment—a technique that produces few-nanometer-thick rare-earth-doped $Y_2O_3$ films with atomic scale thickness control and emission quality as in bulk crystals (see Supplementary Note 1, Supplementary Figs. 1 and 2, and Supplementary Table 1). The results we will show are obtained with a sample containing an erbium layer of 12 nm thickness, as measured through white light interferometry (see Supplementary Note 2 and Supplementary Figs. 3 and 4 for more sample characterization).

With the aim of dynamically controlling the erbium–graphene interactions through the Fermi energy of graphene $E_F$, we combine a backgate of p-doped silicon with a polymer electrolyte topgate[27] (see Fig. 1). The backgate, with a smaller alternating current (AC) impedance than the topgate, is very suitable for high-frequency modulation. We use it to modulate $E_F$ at high frequencies over a range of ~0.3 eV. On the other hand, the topgate allows Fermi energy tuning over a range >1 eV, which is sufficiently high to launch plasmons in resonance with the photons emitted by erbium, whose energy is $E_{Er} = 0.8$ eV (see Fig. 1a). We use the topgate to provide a high base Fermi energy during high-frequency modulation of the backgate. The Fermi energies induced by both gates are calibrated by Hall measure-ments and resistance measurements (see Supplementary Note 3 and Supplementary Fig. 5). Thus, using our dual-gated, hybrid erbium–graphene system, we can modulate $E_F$, for example, between 0.6 and 0.3 eV, (Fig. 1b). This modulates the system between two regimes, where the erbium emitters decay by transferring energy to graphene, leading to intraband absorption (Fig. 1a) and interband absorption (Fig. 1c), respectively. The intraband regime is mainly associated with plasmon generation in graphene, whereas in the interband regime mainly electron–hole pair creation occurs.

### Emission contrast and decay enhancement.
The intraband and interband absorption regimes can be experimentally distinguished because they are associated with different local densities of optical states (LDOS), leading to distinct decay rates for the emitters interacting with graphene. Our calculations following refs. [28,29] show that the decay-enhancement factor $F_P$ for an emitter at 5 nm from graphene is ~100 in the intraband regime ($E_F = 0.6$ eV) and >1000 in the interband regime ($E_F = 0.3$ eV). In order to observe these regimes experimentally, we excite the erbium ions with a 532-nm laser and detect their emission at 1.54 μm in a scanning confocal microscope (see "Methods"), while varying the Fermi energy using the topgate. Figure 2a shows the measured emission contrast, defined as the emission without graphene (measured by shining the laser outside the graphene channel) divided by the emission with graphene (measured by shining the laser on graphene; see also Supplementary Fig. 3). The excitation power is sufficiently low (fluence of $10^4$ W cm$^{-2}$) to ensure that the ion transition is not saturated (see Supplementary Fig. 4). We indeed identify two different regimes: at low Fermi energies ($E_F <$ 0.4 eV), energy transfer is mostly caused by interband transitions in graphene. Above 0.4 eV, interband transitions become sup-pressed by Pauli blocking, because $E_F > E_{Er}/2$, and intraband

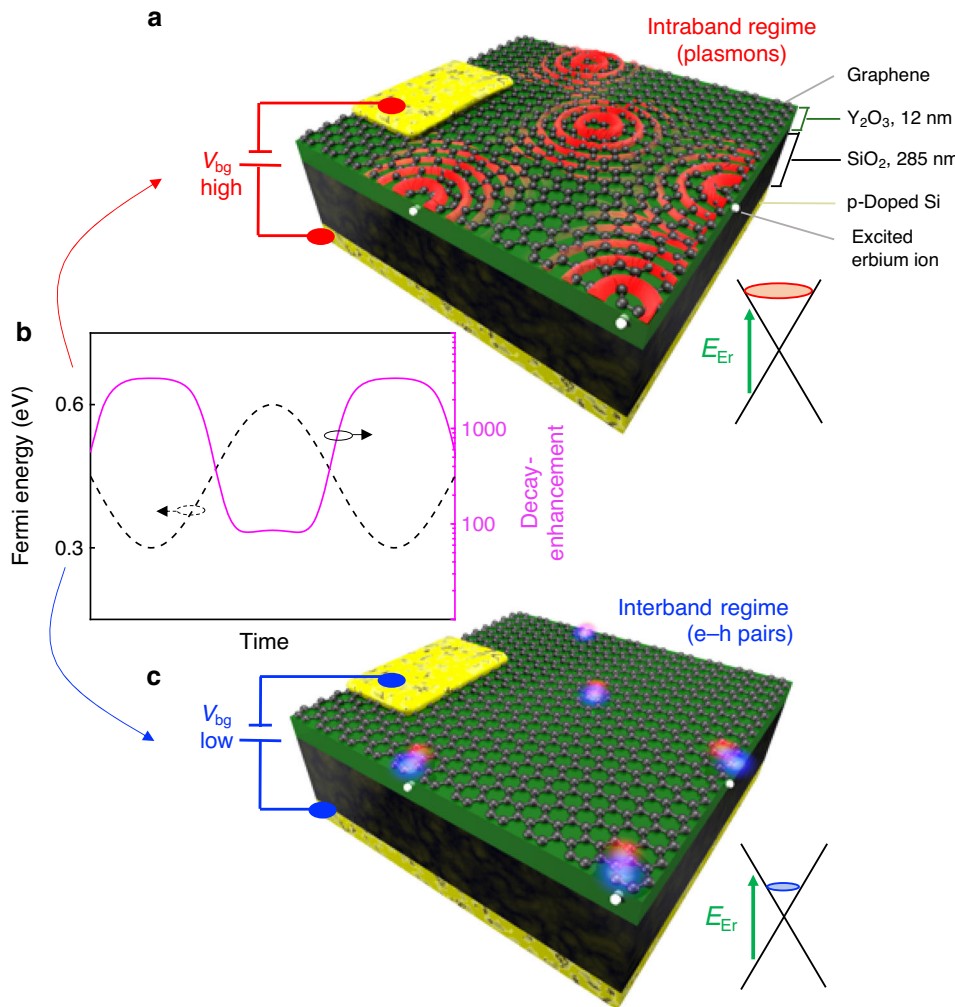

**Fig. 1 Concept of dynamic modulation of hybrid erbium–graphene system. a** Schematic illustration of the hybrid erbium–graphene system when the Fermi energy of graphene is tuned to ~0.6 eV and the erbium–graphene interaction leads to intraband transitions, mainly associated with launching of propagating graphene plasmons (red waves). The system contains, from top to bottom, a monolayer of graphene, a thin film (~12 nm) of $Y_2O_3$ containing erbium ions (white spheres), a 285-nm-thick $SiO_2$ layer, and a $p$-doped silicon backgate. A backgate voltage, $V_{bg}$, is applied between the backgate and a gold electrode in contact with graphene for fast modulation. The $SiO_2$ layer serves as electrical isolation between graphene and the $p$-doped silicon. On top of graphene, there is a transparent topgate made of polymer electrolyte (not shown in the image). **b** Sinusoidal time evolution of the Fermi energy of graphene (dashed black line, left vertical axis) and the corresponding decay-enhancement factor $F_P$ for an erbium emitter located at $z = 5$ nm from graphene (solid purple line, right vertical axis). The modulation of the Fermi energy, calculated following refs. [28,29], leads to a modulation of $F_P$ by more than one order of magnitude—from <100 in the intraband regime to >1000 in the interband regime. **c** Schematic illustration of the hybrid erbium–graphene system for $E_F$ ~ 0.3 eV, which corresponds to interband transitions, mainly creating electron–hole pairs (red–blue spheres).

transitions become the dominant energy decay pathway of the ions. The positive slope of the emission contrast for $E_F > 0.6$ eV is a clear signature of the presence of plasmons. The slope is positive because the decay length of the plasmon field is approximately the same as the plasmon wavelength, $\lambda_{pl}$, which scales linearly with $E_F$. Therefore, as $E_F$ increases, the volume occupied by the plasmon field increases, thus increasing the number of ions interacting with plasmons, and thereby decreasing the amount of emitted light.

To obtain more insight into the dynamics of the erbium–graphene interactions, we measure the erbium emission decay curves. We modulate the excitation laser into pulses and use single-photon counting electronics to create emission histograms (see "Methods"). Figure 2b shows the decay curves measured with the laser shining on graphene for the interband transition regime ($E_F$ ~ 0.2 eV) and for the intraband regime ($E_F$ ~ 0.8 eV), as well as in a region without graphene. We observe faster decay in the intraband regime ($e^{-1}$ time of ~3 ms) and

even faster decay in the interband regime ($e^{-1}$ time of ~1 ms), compared to the decay without graphene ($e^{-1}$ time of ~6 ms), in qualitative agreement with the emission contrast measurements, which show less emission in the interband regime than in the intraband regime.

Strikingly, the decay curves are multi-exponential, with a large negative slope in the beginning of the decay (see inset of Fig. 2b). This multi-exponential behavior stems from the different decay rates $\gamma$ of the emitters, depending on their distance to graphene $z$, since $\gamma$ scales with $z^{-4}$ in the interband regime and with $\exp(-4\pi z/\lambda_{pl})$ in the intraband regime[28,30]. The ions located furthest away from graphene have the lowest energy transfer rates, thus emitting more photons and extending the decay curves to long times. The strongly negative slope in the beginning of the decay curves (with an estimated decay time <100 µs; see inset of Fig. 2b and Supplementary Fig. 10) is due to the high energy transfer rate for small $z$ and indicates the presence of a significant fraction of ions with very strong erbium–graphene interactions.

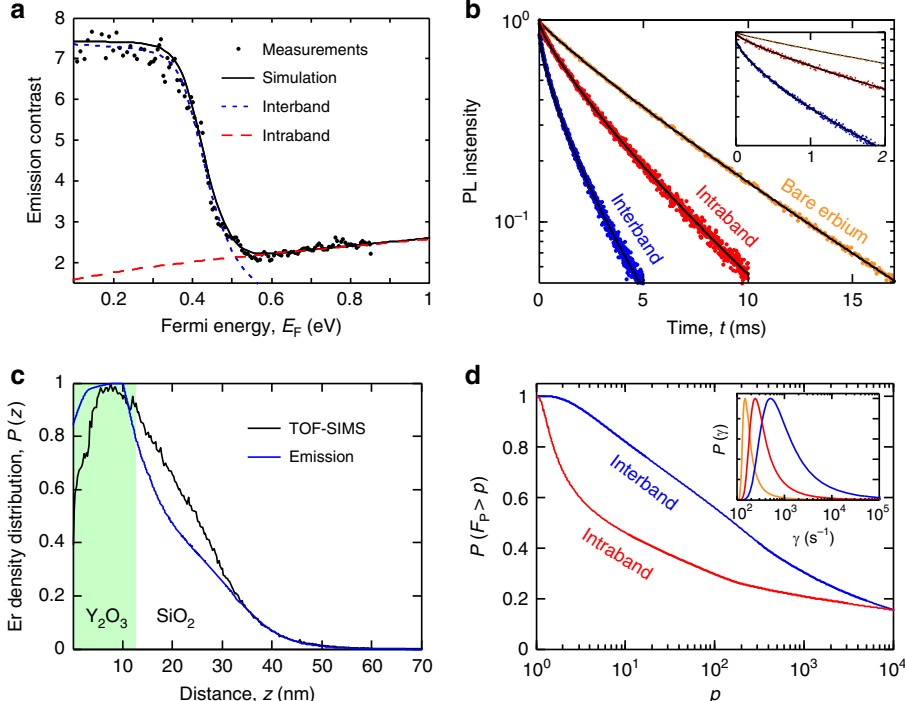

**Fig. 2 Efficient energy flow from erbium emitters to graphene. a** Measured emission contrast, defined as the emission with the excitation laser shining outside graphene divided by the emission with the laser on graphene, as a function of the Fermi energy (black dots). The black solid line represents the calculated emission contrast from $N$ erbium ions located at different distances $z_i$ from graphene (see "Methods"), using the total graphene conductivity. The red (blue) dotted line represents the calculated emission contrast by only considering the conductivity of the intraband (interband) excitation, showing the different microscopic origins of the emitter–graphene interactions at high and low $E_F$. The interband contribution (blue) quickly drops when $E_F > 0.4$ eV, due to Pauli blocking, whereas the intraband contribution (red) steadily rises with $E_F$. **b** Measured decay curves of erbium ions for three cases: graphene in the interband regime ($E_F = 0.3$ eV, blue dots), graphene in the intraband regime ($E_F = 0.8$ eV, red dots), and without graphene (orange dots). The black solid lines are the best-fit stretched-exponential functions. The inset focuses on the beginning of the decay curves, illustrating the multi-exponential behavior. **c** Erbium density distribution (normalized to maximum) vs. erbium–graphene distance obtained from the combined analysis of the emission contrast and the decay curve of the interband regime (blue) and from TOF-SIMS measurements (black). **d** Cumulative distribution of the decay enhancement factor calculated for the interband (blue) and the intraband (red) regimes. These curves show that about 80% (50%) of the ions have $F_P > 10$ for the interband (intraband) regime, and approximately 25% of ions have $F_P > 1000$ for both regimes. The inset shows the total decay rate distributions directly obtained from the multi-exponential fitting of the experimental decay curves (see "Methods"). The colors are the same as in **b**.

Thus the decay curves provide information of the $z$-dependence of the interactions, while the emission contrast measurements reflect the overall (distance-integrated) graphene-induced decay-enhancement $F_P$, which is $\sim 7$ ($\sim 2.5$) in the interband (intraband) regime. This indicates that an overall fraction of $\eta \approx (F_P - 1)/F_P \approx 85\%$ (60%) of the energy of excited erbium emitters flows to interband (intraband) transitions in graphene. We note that we have reproduced these decay curves with multiple erbium–graphene samples (see Supplementary Note 4 and Supplementary Fig. 9).

**Quantifying erbium–graphene interaction.** Given the strong $z$-dependence of the emitter–graphene interaction, it is crucial to determine the distribution of erbium ions in order to quantitatively understand the energy transfer efficiency of the different erbium ions in the nanolayer. We analyze the emission contrast and the decay curves (Fig. 2a, b) together because they provide complementary information. The decay curves provide with high accuracy the distribution of ions with relatively low $F_P$ factors (large $z$), as these are the ions that emit the largest number of photons during lifetime measurements. On the other hand, the emission contrast measurements of Fig. 2a reveal more accurately the effect of the ions with relatively large $F_P$ (small $z$), as these are the ions that transfer the highest fraction of their energy to

graphene, leading to the largest decrease in emission. Together, the decay curves and emission contrast measurements yield the density of erbium ions as a function of $z$ (see Fig. 2c).

In detail, we extract the distribution for ions with $z > 7$ nm directly from the decay curves of Fig. 2b, by describing each multi-exponential decay curve by a continuous sum of exponentially decaying functions, whose probability amplitudes are given by the decay-rate distribution $P(\gamma)$. In terms of the ion dynamics, the distribution $P(\gamma)$ can be interpreted as the likelihood that a given ion decays with a certain decay rate $\gamma$. The inset of Fig. 2d clearly shows that the decay-rate distribution $P(\gamma)$ is shifted toward higher values of $\gamma$ in the regions of the device with graphene, indicating that the decay rate increases due to energy transfer to graphene. From the analysis of these decay-rate distributions, we extract the distribution of decay-enhancement factors $P(F_P)$, following the numerical procedure described in the "Methods" section. Then we convert $P(F_P)$ into the distribution of erbium–graphene distances $P(z)$, i.e., the density distribution, by using the theoretical relation between $F_P$ and the emitter–graphene distance $z$.

It turns out that the distributions $P(F_P)$ and $P(z)$ obtained from the decay curves are accurate up to $F_P = 10^3$, corresponding to $z \gtrsim 7$ nm. For higher values of $F_P$, the decay is so fast and has such a small amplitude that we cannot resolve it experimentally in our decay curves (see Supplementary Note 5). However, we can obtain the distribution of the ions with $z < 7$ nm using the

emission contrast measurements of Fig. 2a. For this, we use a computational model of $N$ ions at different distances from graphene, $z_i$ ($i = 1, ..., N$), and find the ion density distribution $P(z)$ that best reproduces the measurements of Fig. 2a, b (see "Methods"). This is how we obtain the ion density distribution in Fig. 2c.

We compare the ion distribution extracted from optical measurements $P(z)$ with the density distribution measured by means of time-of-flight secondary ion mass spectrometry (see "Methods" and Supplementary Note 6). The similarity between the two density distributions confirms the validity of our analysis. Interestingly, our results indicate that some ions have diffused from the $Y_2O_3$ layer into the underlying $SiO_2$ layer. This has likely occurred during the annealing post-treatment of the films. These diffused ions interact less strongly with graphene and lead to the moderate overall emission contrast we observe in Fig. 2a.

Importantly, our analysis of the experimental data of Fig. 2a, b provides evidence of very strong emitter–graphene interactions at the shortest distances. Figure 2d shows the calculated cumulative distribution of $P(F_P)$,

$$P(F_P > p) = \int_p^\infty P(F_P) dF_P, \tag{1}$$

which describes the probability that an ion has a decay-enhancement factor larger than $p$. In this way, the cumulative distribution $P(F_P > p)$ provides the fraction of ions with $F_P > p$. Using Eq. (1), we find that about 80% (50%) of the ions have decay enhancement factor $F_P > 10$ for the interband (intraband) regime, and approximately 25% of ions have $F_P > 1000$ for both regimes, which means that >99.9% of the energy from these ions flows to graphene.

**Fast electrical modulation of near-field interactions**. Having established the occurrence of highly efficient erbium–graphene interactions in our system, we now demonstrate dynamic control of these interactions on a time scale that is much shorter than the emitter's lifetime of ~10 ms[31]. We induce a fast temporal variation in the LDOS experienced by the emitters by modulating the Fermi energy of graphene. To this end, we apply an AC voltage to the backgate and verify the effect of this modulation on the excited state populations of erbium by measuring the temporal variations of photon emission using a single-photon counting set-up. In these experiments, we keep the excitation laser power constant. In Fig. 3, we modulate the Fermi energy between 0.3 and 0.6 eV at different modulation frequencies $f_{mod}$ between 20 Hz and 300 kHz. This corresponds to a modulation of the erbium decay pathway between interband and intraband transitions, as in Fig. 1. In these measurements, we thus establish control over the timing of plasmon launching from erbium ions down to the microsecond range, which is remarkable for emitters with millisecond natural lifetime. We verified that there is no dynamic response outside graphene and that a backgate voltage of <10 V is sufficient for complete modulation between the two emitter–graphene interaction regimes (see Supplementary Fig. 6).

As $f_{mod}$ increases and becomes higher than the emitter decay rate, the internal dynamics of the ions are not able to follow the temporal variations of the environment. This results in a gradual delay of the maximum and minimum of the time-dependent emission with respect to the sinusoidal oscillation of the Fermi energy and the reduction of the emission modulation amplitude, which depends on $1/f_{mod}$ whenever $f_{mod} \gg \gamma$. Interestingly, these modulation frequencies surpass not only the decay rate of the ions but also the quantum coherence decay rate of erbium in $Y_2O_3$ (11 kHz; measured at 2.5 K in a bulk ceramic sample and under a small external magnetic field of 0.65 T)[32].

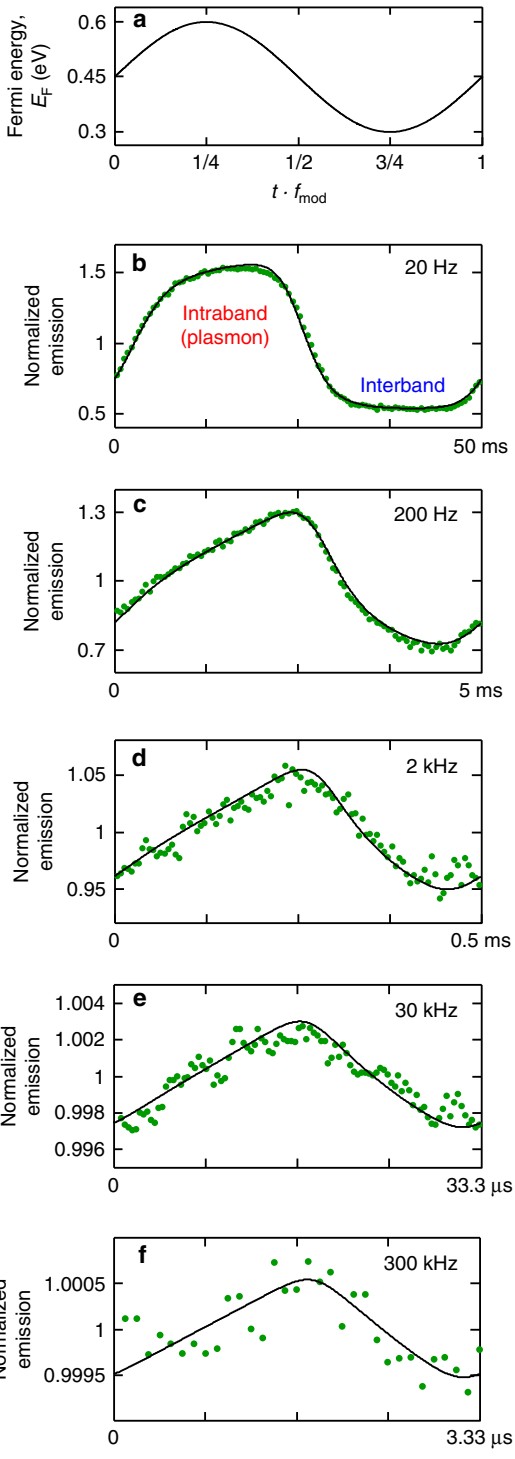

**Fig. 3 Dynamic modulation between interband and intraband regimes. a** Fermi energy as a function of time $t$ using a sinusoidal function of frequency $f_{mod}$ (schematic). The slow topgate is used to tune to $E_F \sim 0.45$ eV, whereas the fast backgate provides the modulation between 0.3 and 0.6 eV. **b–f** Time-resolved photon emission while the Fermi energy is modulated at different frequencies between 20 Hz and 300 kHz (green dots). Every time-modulated emission measurement is normalized to its mean value, so that the modulation does not depend on the excitation laser power or the photon collection efficiency. The black solid curves show the dynamic simulations of $N$ ions located at the distribution of distances from graphene $P(z)$, obtained from the emission contrast measurements and decay curves (see "Methods"). Note that **f** shows modulation on a time scale of microseconds, whereas the radiative lifetime of erbium ions in $Y_2O_3$ is ~10 ms, a difference of 4 orders of magnitude.

In a second series of measurements, shown in Fig. 4, we modulate the Fermi energy between 0.7 and 0.9 eV. Here we apply a higher voltage to the polymer electrolyte topgate than in the previous modulation experiment, while modulating the backgate again with a voltage of <10 V. In this situation, the system is always in the intraband regime, where plasmon launching is the dominant energy decay pathway, and we therefore modulate the strength of the emitter–plasmon interaction. Note that an increase in backgate voltage now leads to a decrease in emission, because stronger emitter–plasmon interaction leads to less emission (see also Supplementary Fig. 7). This is in contrast with the case in Fig. 3, where an increase in backgate voltage leads to a transition from the interband absorption regime to the intraband regime, giving more emission.

We model the temporal modulation of photon emission by simulating the dynamics of N ions using the distribution of ion distances $P(z)$ obtained from the emission contrast measurements and decay curves. For every ion $i$ at distance $z_i$, we numerically solve the rate equation with the time-dependent $F_P$ factor and for the corresponding Fermi energy modulation (see "Methods"). We find good agreement between experimental data and numerical simulations, which do not contain any freely adjustable fit parameters (see "Methods"), adding credibility to our computational approach. We note that, only in the case of Fig. 4e, we observe a larger modulation amplitude in the experiment than in the simulation. We speculate that this could be related to additional (e.g., non-local) effects in the plasmonic local field at very short distances from graphene.

## Discussion

We have demonstrated a material platform that integrates the optical properties of erbium quantum emitters with the strong near-field interactions of graphene, providing the conditions for quantum manipulation and metrological functionalities[33,34]. In particular, our hybrid erbium–graphene platform enables fast temporal control over the strong near-field interactions, thus providing an efficient way to manipulate quantum states in nanoscale solid-state devices by means of conventional electronics. The control over the dynamics of single-photon emitters on a much shorter time scale than their lifetime is an essential ingredient toward observing intriguing effects, such as Dicke phase transitions[22,23], non-linear light–matter interactions at the quantum level[35], and multi-particle entanglement generation[24]. In addition, such a fast control over the near-field interactions will expand existing functionalities of plasmonic nanodevices integrating graphene waveguides and cavities[28,29,36–38]. It enables the control of the waveform of photons and plasmons emitted into a guided mode[21,39], which is a required capability for distributed quantum systems. It furthermore facilitates the connection of optical transistors[40,41] based on quantum emitters in integrated plasmonic circuits, with the possibility to attenuate or amplify the plasmon emission by means of a gate voltage. These promising applications will be boosted by the prospects of coupling plasmonic modes to the far field through optical nanoantennas or through optical waveguides[42], thus enabling controlled photon emission enhancement for optical communications. Overall, these prospects will stimulate the development of emitter–graphene interfaces as a building block for hybrid systems with applications in optoelectronic quantum technologies.

## Methods

**Fabrication of erbium–graphene hybrid devices.** The erbium–yttria thin-film depositions are carried out by ALD with a commercial reactor using conventional β-diketonate precursors: Er(tmhd)3 and Y(tmhd)3. The precursors are held at 160 °C and delivered using $N_2$ as a carrier gas and $O_3$ as an oxidizing agent. The

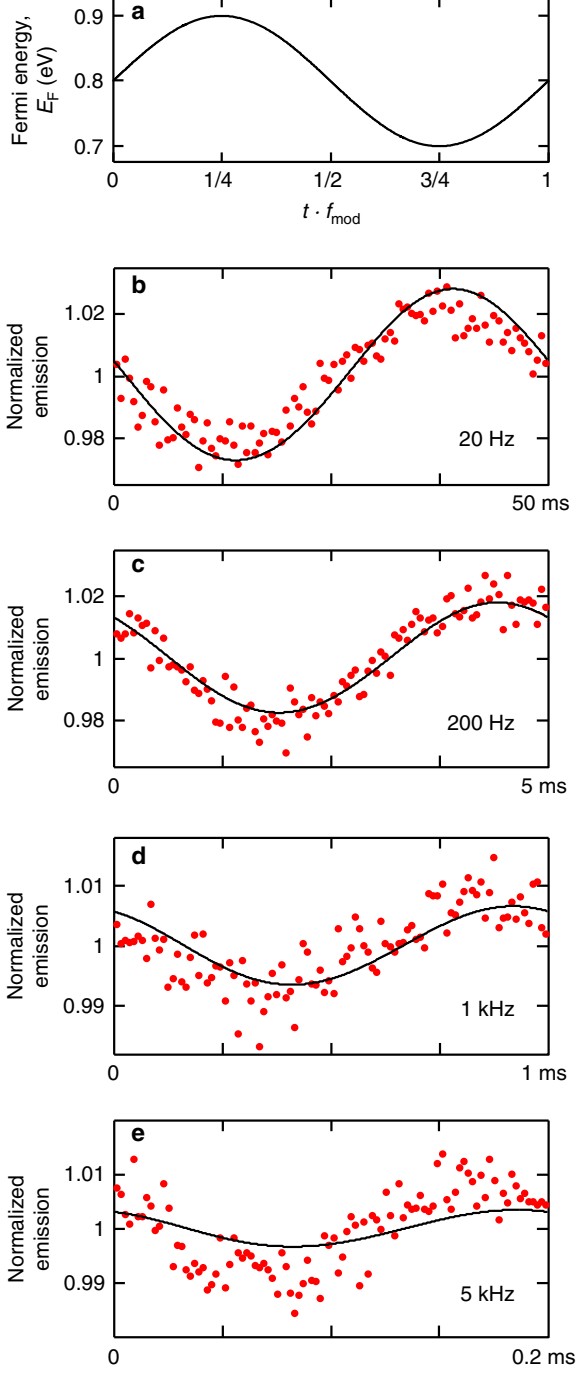

**Fig. 4 Dynamic modulation within the intraband regime. a** Fermi energy as a function of time $t$ using a sinusoidal function of frequency $f_{mod}$ (schematic). The slow topgate is used to tune to $E_F$ ≈ 0.8 eV, whereas the fast backgate provides the modulation between 0.7 and 0.9 eV, thus modulating the interaction strength within the intraband regime. **b–e** Time-resolved photon emission while the Fermi energy is modulated at different frequencies between 20 Hz and 5 kHz (red dots). Every time-modulated emission measurement is normalized to its mean value, so that the modulation does not depend on the excitation laser power or the photon collection efficiency. The black solid curves show the dynamic simulations of N ions located at the distribution of distances from graphene $P(z)$ obtained from the emission contrast measurements and decay curves (see "Methods"). Note that the modulated emission is out of phase with the modulating Fermi energy, because a higher Fermi energy gives lower emission in the intraband regime.

precursors are flown sequentially with 3-s injection time into the thermalized deposition chamber at 350 °C. The number of cycles is adjusted in order to obtain the desired thickness (circa 12 nm here). The films are annealed at high temperature (950 °C) for 2 h in air prior to measurement in order to improve crystallinity. More details about the optimization of the growth procedure are discussed in Supplementary Note 1. We use a moderate erbium concentration of 2% in order to avoid possible non-radiative decay channels induced by erbium–erbium interactions. Single-layer graphene, grown by chemical vapor deposition on copper, is transferred directly onto the surface of the 12-nm-thick $Y_2O_3$:$Er^{3+}$ layer using the standard wet transfer method. Graphene is patterned into a Hall-bar geometry by laser writing optical lithography and subsequent reactive ion etching (see Supplementary Fig. 3). The whole surface is covered with a transparent polymer electrolyte that serves as topgate. The electrolyte is made of polyethylene oxide and $LiClO_4$ with 8:1 weight ratio in a solution of methanol[27]. The device contains six electrical contacts to graphene and two electrical contacts to apply the topgate voltage $V_{tg}$ to the polymer electrolyte. The electrical contacts consist of a 50-nm-thick gold layer deposited on a 5-nm-thick chromium sticking layer and patterned by laser writing optical lithography. There is a 50-nm-thick $SiO_2$ protection layer between the gold electrodes and the electrolyte in order to isolate the gold electrodes from the polymer electrolyte.

**Experimental set-up for optical measurements**. The experiments are carried out in a home-built scanning confocal microscope set-up, with the sample mounted inside a vacuum chamber at a pressure of 5–10 mbar for optimal operation of the polymer electrolyte gate. The optical set-up uses an infrared objective (Olympus LCPLN-50X-IR, numerical aperture 0.65), which provides a spatial resolution of ~1 μm. A focused 532-nm laser beam, with typically a power of ~0.2 mW at the sample, excites the ions into the short-lived state $^2H_{11/2}$, from which the erbium population rapidly decays into the metastable first excited state $^4I_{13/2}$ via non-radiative multiphonon emission. The emission of the $^4I_{13/2} \rightarrow {}^4I_{15/2}$ transition at the characteristic wavelength 1.54 μm is collected, spectrally filtered with a narrow bandpass filter (Thorlabs FB1550-40), and directed into a near-infrared single-photon detector (ID Quantique id210) with very low level of dark counts (~9 Hz). For time-resolved measurements, emission histograms are obtained using photon-counting electronics (PicoHarp 300) in time-tagged time-resolved acquisition mode, recording the arrival times of all photons. During lifetime measurements, the excitation laser intensity is modulated into square pulses by switching on and off the signal of an acousto-optic modulator. All measurements were carried out at room temperature. We verified that, during measurements that can take up to several hours, the Fermi energy did not vary significantly (see Supplementary Fig. 8).

**Decay-rate distributions from the decay curves**. The decay rate in the regions of the device without graphene is $\gamma_{Er} = \gamma_{ed} + \gamma_{md} + \gamma_{nr}$, where $\gamma_{ed} \sim 75$ Hz and $\gamma_{md} \sim 50$ Hz correspond to the electric and magnetic dipole moments, respectively[31], and $\gamma_{nr}$ represents the non-radiative decay channels of the erbium thin film. In the regions with graphene, the total decay rate is $\gamma = \gamma_{Er} + \gamma_{gr}$, where $\gamma_{gr}$ is the rate of energy transfer to graphene. Every experimental decay curve, $n(t)$, is described as a continuous sum of exponential decays, $n(t) = \int_0^\infty \frac{P(\gamma)}{\gamma} e^{-\gamma t} d\gamma$, where $P(\gamma)$ is the probability distribution that describes the likelihood that a given ion decays at a certain rate $\gamma$. Here we have considered that the excited erbium population, and thus the emission, is inversely proportional to the decay rate. The integral has the form of a Laplace transformation, so we can extract $P(\gamma)$ by inverse Laplace transformation using the numerical techniques of ref. [43] (see the inset of Fig. 2d). Next, we calculate the energy transfer rate distribution $P_{gr}(\gamma_{gr})$ by numerically solving the convolution equation, $P(\gamma) = \int_0^\infty P_{Er}(\gamma - \gamma_{gr}) P_{gr}(\gamma_{gr}) d\gamma_{gr}$. Here $P_{Er}(\gamma_{Er})$ denotes the decay-rate distribution in the regions of the device without graphene (we write the subindex Er to indicate that the distribution only includes the intrinsic decay mechanisms of the erbium film). In doing the deconvolution, we filter out the small effect of the undesired non-radiative decay channels $\gamma_{nr}$, thus obtaining the pure contribution of the erbium–graphene interactions, $\gamma_{gr}$. The distribution $P_{gr}(\gamma_{gr})$ can be easily converted into the distribution of decay-enhancement factors, $P(F_P)$, by using the relation $\gamma_{gr} = (F_P - 1)\gamma_{ed}$. Next, we translate $P(F_P)$ into the density distribution, $P(z) = P(F_P) dF_P/dz$. For this, the decay-enhancement factor as a function of distance, $F_P(z)$, is calculated as in refs. [28,29], where the response of graphene to the localized field of the emitter is simulated using the optical conductivity of the Kubo model, with a typical momentum scattering time of $\tau_{sc} = 50$ fs and a refractive index of 1.8 for $Y_2O_3$ and 1.4 for the electrolyte (see Supplementary Fig. 9). The distributions obtained from the decay curves are very accurate for the lowest decay-rate enhancements factors, $F_P \lesssim 1000$. This cutoff is determined from the accuracy of the numerical inverse Laplace transform (see Supplementary Note 5). Ions with larger decay rates emit so few photons that they barely affect the slope of the decay curves, and their energy-transfer rates have to be investigated by using the emission contrast measurements. We have repeated the whole procedure using the decay curves of different Fermi energies (0.3 and 0.8 eV), and the results are practically the same.

**Decay-rate distributions using the $N$-ion model**. We use a computational model of $N$ ions located at different distances from graphene, $z_i$ ($i = 1, ..., N$). The positions $z_i \gtrsim 7$ nm are obtained by discretization of the density distribution $P(z)$ calculated from the decay curves. The positions $z_i \lesssim 7$ nm are free parameters that we vary to find the distribution of distances, $\{z_i\}_{i=1,...,N}$, that best reproduces the emission contrast measurements of Fig. 2a. This variational procedure assumes that the density distribution is smooth. The calculations are accomplished by considering that the emission from every ion is proportional to its excited-state population, which in turn is proportional to $1/(F_P\gamma_{ed} + \gamma_{md} + \gamma_{nr})$, where the theoretical decay-enhancement factor $F_P(z_i)$ is computed from the methods of refs. [28,29] (see above). Here we assume that $\gamma_{nr} = 10$ Hz for all ions (see Supplementary Note 1). The discrete distribution $\{z_i\}_{i=1,...,N}$ is converted into a continuous distribution, $P(z)$, and vice versa, by integration and discretization, respectively. We typically use $N = 50$, where the density distribution already nicely converges.

**Simulation of the dynamic response to gate modulation**. We simulate the quantum-state population dynamics for every ion $i$ of our $N$-ion model by numerically solving the rate equations[44],

$$\frac{dN_e^{(i)}}{dt} = -N_e^{(i)}\left[F_P^{(i)}\gamma_{ed} + \gamma_{md} + \gamma_{nr}\right] + N_g^{(i)}\gamma_{exc}, \tag{2}$$

$$N_g^{(i)} + N_e^{(i)} = 1, \tag{3}$$

where $N_g^{(i)}$ and $N_e^{(i)}$ are, respectively, the populations in ground and excited states, $F_P^{(i)}$ is the time-dependent decay-enhancement factor at the position $z_i$, and $\gamma_{exc}$ is the excitation rate, which depends on the excitation laser power. The results of the simulations do not depend on $\gamma_{exc}$ since every measurement is normalized to its mean value, although we need to assume a certain value of $\gamma_{exc}$ to do the computation. To calculate $F_P^{(i)}$, we first obtain the oscillating energy $E_F$ that corresponds to the applied gate voltages using the calibration described in Supplementary Note 3. The Fermi energy amplitude as a function of the back-gate voltage amplitude is $\Delta E_F = B\Delta V_{bg}$, where $B = 15$ eV mV$^{-1}$ (13 eV mV$^{-1}$) in the experiments of Fig. 3 (Fig. 4). We then convert $E_F$ into $F_P$ using the methods of refs. [28,29] (see above).

**Time-of-flight secondary ion mass spectrometry (ToF-SIMS)**. We use a dual-beam ToF-SIMS spectrometer (IONTOF GmbH, Műnster, Germany) to measure the density profile of the yttrium and erbium ions at a sufficiently low primary ion dose density to keep static conditions. The spectrometer was operated at a pressure of $10^{-9}$ mbar. A pulsed 25 kV Bi$^+$ primary ion beam delivering 1 pA over a 100 × 100 μm$^2$ area is used to etch the chemical species from the surface. The masses of the removed chemical species are determined by ToF-MS. The sputtering of the surface was done using a 2-keV Cs$^+$ sputter gun giving a 100-nA target current over a 300 × 300 μm$^2$ area. The interlacing between Bi$^+$ and Cs$^+$ guns allows to record TOF-SIMS depth profiles. We used the profile of the removed YO$^-$ particles to extract the Er$^{3+}$ density distribution. This approximation is justified since the diffusion coefficients of yttrium and erbium are practically the same (see Supplementary Fig. 11).

## Data availability
The data that support the findings of this study are available from the corresponding authors upon reasonable request.

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

## Acknowledgements

We thank Javier García de Abajo, David Hunger, and Dmitri Efetov for discussions and David Saleta, Xiaoyu Jia, Juan Sierra, Marius Costache, and Sergio Valenzuela for assistance with the Hall measurements. K.-J.T. acknowledges funding from the European Union's Horizon 2020 research and innovation program under Grant Agreement No. 804349. ICN2 was supported by the Severo Ochoa program from Spanish MINECO (Grant No. SEV-2017-0706). This project has received funding from the European Union's Horizon 2020 research and innovation program under grant agreement No. 712721 (NanOQTech). F.H.L.K. acknowledges financial support from the Government of Catalonia trough the SGR grant, and from the Spanish Ministry of Economy and Competitiveness, through the "Severo Ochoa" Programme for Centres of Excellence in R&D (SEV-2015-0522), and Explora Ciencia FIS2017-91599-EXP. F.H.L.K. also acknowledges support by Fundacio Cellex Barcelona, Generalitat de Catalunya through the CERCA program, and the Mineco grants Plan Nacional (FIS2016-81044-P) and the Agency for Management of University and Research Grants (AGAUR) 2017 SGR 1656. Furthermore, the research leading to these results has received funding from the European Union's Horizon 2020 under Grant Agreements No. 785219 (Core2) and No. 881603 (Core3) Graphene Flagship, and No. 820378 (Quantum Flagship). This work was supported by the ERC TOPONANOP under Grant Agreement No. 726001.

## Author contributions

K.-J.T., F.H.L.K., H.d.R., P.G., and A.F. designed the research project. K.-J.T. planned and coordinated the experiments. D.C. carried out the electro-optical measurements with help from K.-J.T. and A.R.-P. A.F., M.S., and A.T. optimized and performed the fabrication of erbium-doped substrates. K.S. fabricated the graphene devices with help from A.R-P, D.C., and K.-J.T. A.S., P.M., M.S., and A.F. performed the ToF-SIMS measurements. D.C. did the numerical simulations and fitting of experimental data. D.C., K.-J.T. and F.H.L.K. interpreted the data. D.C. and K.-J.T. wrote the paper with input from all authors.

## Competing interests

The authors declare no competing interests.
