## [Peer Review File · Nature Communications]

REVIEWER COMMENTS

Reviewer #1 (Remarks to the Author):

This manuscript shows that graphene can be used to electrically modulate the decay rate of nearby emitters (a few nm) by modulating the concentration of carriers in the graphene sheet. The authors clearly demonstrate two different decay channels (plasmon generation, e-h pair generation) by tuning the concentration, or Fermi energy, of the graphene in a 'sweet spot' where half the cycle is governed by interband transitions (e-h pair generation) and the other half of the cycle is governed by intraband transitions (plasmon generation). The authors exploit the high bandwidth of the AC response in graphene to demonstrate that this near-field interaction can be operated in the kHz regime and ingeniously use a combination of Si backgate and polymer top gate to be able to reach this regime without applying large AC voltages while having transparency to optical experiments. An extensive and thorough theoretical analysis corroborates their results. This is not surprising given the well established expertise of the authors in the field. The paper is very clearly written and the results are very convincing.

I believe that this paper will be of great interest to the general physics community (plasmonics, 2D materials, optics, electronics) and given the remarkable high quality of the paper and the clarity of the results, I would recommend publication in Nature Communications. Nevertheless, the manuscript can be improved by addressing the following points:

Abstract:

- 'of a significant fraction of the erbium emitters': what does significant mean? After reading the paper I see it is 25%. Maybe better to use this number in the abstract?

Page 2:

1st paragraph:

-Going briefly through the references presented, it seems to me (although I might be wrong) that reference 11 does not 'modulate the effective refractive index of the cavity' because I fail to recognize an optical cavity. It does however modulate the strength of the interaction at high speed by using graphene. It seems to me that it is a very pertinent reference, but misleading in how it is presented.

-2nd paragraph: The authors attempt to compare field confinement of graphene plasmons with noble metal plasmons, but offer numbers only for graphene plasmons: 100 times smaller than free-space photons. Could they also include the metric for noble metal plasmons?

-2nd paragraph: In the manuscript the 10^6 mode volume confinement is referenced with ref 14. I can't find this metric in the reference. Could the authors help me understand how they obtain this number from the reference?

fig 1:

-I am confused by panel b. It looks as if when the Fermi energy is 0.3eV, then plasmons are launched (intraband transitions) and when $E_f=0.6\text{eV}$ e-h pairs are generated (interband). Surely this is not the case. I realize now that I was wrong, but it took me a while. The fig is correct, but it is confusing. Additionally, for a general audience it would be suitable to show the conical band structure with the two types of transitions.

fig 2:

-fig 2b shows the lifetime in the interband regime at $E_f = 0.2\text{eV}$, however in panel a measurements stop at $E_f=0.3\text{eV}$. Although fig 2a is already convincing, if measurements are available to show the plateau below $E_f=0.3\text{V}$ it would make it even more complete.

-Inset of fig 2b is not mentioned in the caption.

Page 5:

-2nd paragraph: at the end of the paragraph the authors show the % of energy flowing into the different quasiparticles. Just for the form, it would be easier to follow if the authors referred to either interband (intraband) or e-h (plasmons). And this could also be corrected throughout the paper (in the figures for example).

Page6:

-2nd paragraph: I have difficulty following how to obtain the percentage of ions having a certain purcell factor F_p from fig 2d. Could the authors explain it better please? Also, what is the x-axis 'p' in the figure?

Page 7:

-2nd paragraph: define ' γ '. Just for the form, we know it's the decay rate.

Page 8:

-1st paragraph: The authors state that for fig 4e, the modulation amplitude is higher in exp than in simulation. Could this be a margin of error issue (noting that $1/f$ noise could increase this error) instead of non-local effects in the plasmonic local field?

Reviewer #2 (Remarks to the Author):

In this paper, the authors realized fast modulation of the strong near interactions between a nanolayer of erbium emitters and graphene by in-situ electrical tuning the Fermi energy of graphene. In specific, strong interaction with over 1000-fold increased decay rate of the erbium emitters and interactions with frequencies up to 300 kHz were reached. In my opinion, dynamical tuning the strong emitter-graphene interaction is important, which can find applications for integrated quantum technologies. This work is a following one by the same authors (Nat. Phys. 11, 281 (2015)), in which fast dynamical modulations is considered here. The work is interesting, the physics is clean and the results are well presented, I think it can be acceptable as long as following issues are considered.

1. It is difficult to understand the meaning of the inset in Fig.2. It seems that the z dependent decay rate should be discussed here, but it is unclear because that there is a lack of clear description of coordinates and enough discussion in the main text.
2. Overall the paper, the emission is dominated by the isolated erbium emitter with specific distributions, however, collective interactions between them may be another important effect in this extreme condition. Could the authors comment more about if there are collective effects for the erbium emitters?
3. It is still unclear the ultimate limit for this kind of fast modulations. It is clear from Fig. (3) that the decay of the erbium emitters cannot follow the speed of the modulation in the high frequencies. I suggest the authors to give more clear consideration in this aspect.
4. I suggest more applications for this modulation can be included in the revised one, especially on how to implement this kind of modulations in a specific quantum device.
5. In Page 4, the authors have stated that there is an increasing slope of the emission contrast for $E_F > 0.6\text{ eV}$. However, I cannot find the phenomenon in the figure.

We have considered all the recommendations made by the Reviewers. In the revised version of the manuscript, we have included additional data as well as more detailed explanations to clarify the issues raised by the Reviewers. Below we provide our responses to all comments of the Reviewers, and indicate the implemented changes (in blue).

Furthermore, following the journal formatting style, we have removed references from the abstract, and modified the reference list accordingly.

Reviewer 1:

Comment 1: *Abstract: -'of a significant fraction of the erbium emitters': what does significant mean? After reading the paper I see it is 25%. Maybe better to use this number in the abstract?*

⇒ Following the Reviewer's comment, we have written this number in the abstract (Page 1).

Comment 2: *-Going briefly through the references presented, it seems to me (although I might be wrong) that reference 11 does not 'modulate the effective refractive index of the cavity' because I fail to recognize an optical cavity. It does however modulate the strength of the interaction at high speed by using graphene. It seems to me that it is a very pertinent reference, but misleading in how it is presented.*

⇒ We have clarified the description of this reference (Page 2, paragraph 1).

Comment 3: *-2nd paragraph: The authors attempt to compare field confinement of graphene plasmons with noble metal plasmons, but offer numbers only for graphene plasmons: 100 times smaller than free-space photons. Could they also include the metric for noble metal plasmons?.*

⇒ We have included that for noble metal thin films the plasmon wavelength is at most 3 times smaller than the free-space photon wavelength (Page 2, paragraph 2). We have also cited the book of Principles of Nano-Optics by L. Novotny, whose 12th chapter is dedicated to surface plasmons.

Comment 4: *-2nd paragraph: In the manuscript the 10^6 mode volume confinement is referenced with ref 14. I can't find this metric in the reference. Could the authors help me understand how they obtain this number from the reference?*

Answer: This metric is the cubic power of (λ_0 / λ_{pl}). We write it in the paper to remark that the electromagnetic confinement applies to the three dimensions of space: parallel to the graphene monolayer the plasmon propagates with reduced wavelength λ_{pl} , and normal to the graphene monolayer the plasmon field decays as $\exp(-4\pi z/\lambda_{pl})$.

⇒ We have clarified that 10^6 is the cubic power of λ_0 / λ_{pl} , (Page 2, paragraph 2).

Comment 5: *fig 1: -I am confused by panel b. It looks as if when the Fermi energy is 0.3eV, then plasmons are launched (intraband transitions) and when $E_f=0.6\text{eV}$ e-h pairs are generated (interband). Surely this is not the case. I realize now that I was wrong, but it took me a while. The fig is correct, but it is confusing. Additionally, for a general audience it would be suitable to show the conical band structure with the two types of transitions.*

⇒ We have clarified this figure, adding Dirac cones, as suggested by the Reviewer (Page 18).

Comment 6: *fig 2: -fig 2b shows the lifetime in the interband regime at $E_f = 0.2\text{eV}$, however in panel a measurements stop at $E_f=0.3\text{eV}$. Although fig 2a is already convincing, if measurements are available to show the plateau below $E_f=0.3\text{V}$ it would make it even more complete.*

⇒ Following the Reviewer's suggestion, we have included an additional series of measurements for the range of Fermi energies that correspond to the plateau below $E_F = 0.3\text{ eV}$ (Page 19).

Comment 7: *-Inset of fig 2b is not mentioned in the caption.*

⇒ We have written a description of the inset of Fig 2b in the caption (Page 19).

Comment 8: *Page 5: -2nd paragraph: at the end of the paragraph the authors show the % of energy flowing into the different quasiparticles. Just for the form, it would be easier to follow if the authors referred to either interband (intraband) or e-h (plasmons). And this could also be corrected throughout the paper (in the figures for example).*

⇒ We have followed the Reviewer's suggestion and used the terminology intraband/interband transitions throughout the paper, including the figures. Also, for the sake of clarity we explain that the intraband regime is mainly associated with plasmons, whereas the interband regime is mainly associated with e-h pair creation (Page 4, paragraph 1).

Comment 9: *Page6: -2nd paragraph: I have difficulty following how to obtain the percentage of ions having a certain purcell factor F_p from fig 2d. Could the authors explain it better please? Also, what is the x-axis 'p' in the figure?*

Answer: The percentage of ions whose F_P factor is larger than a certain value p is obtained from the cumulative probability distribution $P(F_P > p)$, which is represented in Fig 2d. The reviewer is right to point out that the definition of $P(F_P > p)$ was missing in the first version of the manuscript. That made it difficult to understand how to obtain the percentage of ions from the curves of Fig 2d.

⇒ We have included the definition of $P(F_P > p)$ on Page 7, formula 1, where p is the minimum value of the integral.

Comment 10: Page 7: -2nd paragraph: define 'gamma'. Just for the form, we know it's the decay rate.

⇒ We have included the definition of gamma (Page 6, paragraph 2).

Comment 11: Page 8: -1st paragraph: The authors state that for fig 4e, the modulation amplitude is higher in exp than in simulation. Could this be a margin of error issue (noting that 1/f noise could increase this error) instead of non-local effects in the plasmonic local field?

Answer: The reviewer is raising an interesting question because fig 4e shows the only measurement in which the amplitude of the measured signal is clearly larger than that obtained from the theoretical model. However, the noise does not seem to be the reason for the disagreement between theory and experiment because most the experimental points of Fig 4e are clearly located at larger amplitudes than the theoretical curve. Given the good agreement between experiments and calculations in the other figures, we think that the disagreement in Fig 4e may be due to a physical aspect of the plasmon field that comes up only at high Fermi energies and at high frequencies, and that our theoretical model does not consider. Our hypotheses about the disagreement of Fig 4e are speculative and could be a good motivation for future studies.

The noise actually comes from the Poissonian statistics of the photon counts and the 1/f noise plays a minor role. The signal-to-noise decreases with f_{mod} just because the amplitude of the oscillating signal decreases as f_{mod} increases.

Reviewer 2:

1. It is difficult to understand the meaning of the inset in Fig.2. It seems that the z dependent decay rate should be discussed here, but it is unclear because there is a lack of clear description of coordinates and enough discussion in the main text.

We have included more information in the paper in order to complete the description of Fig 2. In particular:

⇒ To clarify the meaning of the inset of Fig. 2, we have made the following modifications. On page 6 (second paragraph), we have improved the description of the distributions $P(\gamma)$ shown in the inset. On page 11 (Methods), we have extended the explanations about how the distributions $P(\gamma)$ are obtained. We have also included a comment in the caption of Fig. 2 to refer to the inset.

⇒ To discuss the z-dependent decay rate, we have included the following information. In Fig S9, we have included the theoretical functions $F_P(z)$ as well as the erbium-graphene separations z_i used in our N-ion model. In section D of the supplement, we have extended the discussion about the z dependent decay rate.

⇒ To describe the coordinates of Fig 2, we have made the following changes. On page 7 (second paragraph), we have written the definition of the cumulative distribution $P(F_P > p)$, including the meaning of p .

2. *Overall the paper, the emission is dominated by the isolated erbium emitter with specific distributions, however, collective interactions between them may be another important effect in this extreme condition. Could the authors comment more about if there are collective effects for the erbium emitters?*

Answer: We agree that collective interactions could represent an important issue. There are two major effects that collective interactions can cause.

-The first one is the energy transfer between the ions. This effect is independent from the erbium-graphene interactions and has no impact on the F_P factors. However, the energy transfer between ions may induce other detrimental effects like energy migration into quenching centers. Our measurements confirm that such detrimental effects are negligible for the moderate erbium concentration of our thin film samples (with 2% erbium ion concentration), as demonstrated by the fact that the intrinsic non-radiative decay rate of the bare substrate is very small. Moreover, the deconvolution described in Methods filters out any sort of non-radiative decay, regardless whether it is caused by erbium-erbium energy transfer or by another intrinsic mechanism. For this reason, we are convinced that energy transfer between erbium ions plays a negligible role.

-The second effect to be considered is collective erbium-graphene interactions. This effect would induce a collective enhancement of the F_P factors, like what happens in the phenomenon of superradiance. However, we can rule out the possibility of collective erbium-graphene interactions because of the low erbium population in the excited state: the number of excited ions inside the plasmon mode volume is less than one.

⇒ To clarify this point, we have written a comment on Page 9 (bottom) of the main text. In the supplement, we have written a comment at the end of section A and a paragraph at the end of section B.

3. *It is still unclear the ultimate limit for this kind of fast modulations. It is clear from Fig. (3) that the decay of the erbium emitters cannot follow the speed of the modulation in the high frequencies. I suggest the authors to give more clear consideration in this aspect.*

Answer: The ultimate limit of this kind of fast modulation is given by two frequencies. The first frequency is the energy-transfer rate γ_{gr} , which determines the maximum modulation frequency that the ion dynamics can follow. For our system, we estimate this to be on the order of 100 MHz. By including resonant plasmonic structures, for example, this could be enhanced even further. The ultimate limit is then given by a second

frequency, namely the maximum modulation frequency of the AC response of graphene, which is tens of GHz.

⇒ We have included the Figure S9a, from which we deduce that the highest energy transfer rates in our system are of the order of 100 MHz. To describe these limits, we have written a paragraph at the end of section D of the supplement.

4. *I suggest more applications for this modulation can be included in the revised one, especially on how to implement this kind of modulations in a specific quantum device.*

Answer: We support the reviewer's affirmation that the dynamical tuning of the strong emitter-graphene interactions can find applications in integrated quantum technologies.

⇒ Following this suggestion, we have included a description of some quantum applications in the conclusion (Page 9), adding some citations to remark the relevance of our findings to the field of quantum plasmonics.

5. *In Pape 4, the authors have stated that there is an increasing slope of the emission contrast for $E_F > 0.6$ eV. However, I cannot find the phenomenon in the figure.*

⇒ We have changed "increasing slope" into "positive slope" in the new version of the paper (Page 4, bottom).

REVIEWERS' COMMENTS:

Reviewer #1 (Remarks to the Author):

The authors have satisfactorily addressed all the points raised in my report.

Reviewer #2 (Remarks to the Author):

In the revised paper, the authors have made great efforts to answer the comments raised by the referees. In my opinion, my previous comments on the drawbacks of the present works have been considered in the revised manuscript, and the quality of the manuscript has been improved a lot. As a result, I recommend its publication in Nat. Commun. as it is.

Dear Editor,

Please find below our response to the Reviewers.

Kind regards,

on behalf of all authors,

Klaas-Jan Tielrooij

Reviewer #1 (Remarks to the Author):

The authors have satisfactorily addressed all the points raised in my report.

⇒ We thank the Reviewer for his/her positive evaluation.

Reviewer #2 (Remarks to the Author):

In the revised paper, the authors have made great efforts to answer the comments raised by the referees. In my opinion, my previous comments on the drawbacks of the present works have been considered in the revised manuscript, and the quality of the manuscript has been improved a lot. As a result, I recommend its publication in Nat. Commun. as it is.

⇒ We thank the Reviewer for his/her positive evaluation.